# A New Species of *Neoscytalidium hylocereum* sp. nov. Causing Canker on Red-Fleshed Dragon Fruit (*Hylocereus polyrhizus*) in Southern Thailand

**DOI:** 10.3390/jof9020197

**Published:** 2023-02-03

**Authors:** Prisana Wonglom, Chaninun Pornsuriya, Anurag Sunpapao

**Affiliations:** 1Faculty of Technology and Community Development, Phatthalung Campus, Thaksin University, Phatthalung 93210, Thailand; 2Agricultural Innovation and Management Division (Pest Management), Faculty of Natural Resources, Prince of Songkla University, Hat Yai District, Songkhla 90110, Thailand

**Keywords:** canker, morphology, molecular identification, pathogenicity test, pitaya

## Abstract

During 2020–2021, cultivated red-fleshed dragon fruit (*Hylocereus polyrhizus*) in Phatthalung province, southern Thailand, was infected with canker disease in all stages of growth. Small, circular, sunken, orange cankers first developed on the cladodes of *H. polyrhizus* and later expanded and became gray scabs with masses of pycnidia. The fungi were isolated using the tissue transplanting method and identified based on the growth of the fungal colony, and the dimensions of the conidia were measured. Their species level was confirmed with the molecular study of multiple DNA sequences, and their pathogenicity was tested using the agar plug method. Morphological characterization and molecular identification of the internal transcribed spacer (ITS), translation elongation factor 1-α (*tef1-α*) and β-tubulin (*tub*) sequences revealed the fungal pathogen to be a new species. It was named *Neoscytalidium hylocereum* sp. nov. The biota of the new species, *N*. *hylocereum*, was deposited in Mycobank, and the species was assigned accession number 838004. The pathogenicity test was performed to fulfil Koch’s postulates. *N. hylocereum* showed sunken orange cankers with a mass of conidia similar to those observed in the field. To our knowledge, this is the first report of *H. polyrhizus* as a host of the new species *N. hylocereum* causing stem cankers in Thailand.

## 1. Introduction

Dragon fruit, also known as pitaya, is the fruit of several different cactus species belonging to the family of Cactaceae. The stems of dragon fruit are scandent, sprawling or clambering, and they branch profusely, whereas the fruit is oblong to oval and contains white or red pulp as well as edible black seeds. Dragon fruit is well known for its composition rich in nutrients, fiber, vitamins and minerals [1] and corresponding high medicinal value, with the potential to lower blood cholesterol concentration [2]. Due to its beneficial nutritive properties, this plant is now spreading worldwide for its fruit, especially in Asian countries such as Malaysia, the Philippines, Taiwan, Thailand and Vietnam [3,4]. Three varieties are commonly cultivated, including the red-skinned fruit with white flesh (*Hylocereus undatus*), red-skinned fruit with red flesh (*H. polyrhizus*) and yellow-skinned fruit with white flesh (*H. megalanthus*) [5]. Although the cultivation of dragon fruit is historically common in all parts of Thailand, its cultivation has helped to provide additional income to farmers (Department of Agriculture Extension).

Similar to most other crops, the cultivation of dragon fruit faces problems caused by several fungal pathogens that have been scientifically documented, such as *Bipolaris cactivora*, *Botryosphaeria dothidea*, *Colletotrichum capsici*, *C. gloeosporioides*, *C. truncatum*, *Curvularia lunata*, *Fusarium semitectum*, *Gilbertella persicaria*, *Neoscytalidium dimidiatum* and *Monilinia fructicola* [4,6,7,8,9,10]. In Thailand, dragon fruit is cultivated in all regions—especially the southern region, in which the weather is suitable for plantations. The weather in tropical and subtropical zones is also favorable for pathogen germination and spread [11,12]. According to the pest list database, the identification of diseases in dragon fruit is rarely reported. Recently, flower rot of red-fleshed dragon fruit was reported to be caused by the fungus *Gilbertella persicaria* in southern Thailand [13]. Dy et al. [14] also demonstrated that the fungus *N. dimidiatum* acted as a pathogen, causing stem canker in *H. polyrhizus* in southern Thailand.

However, there are dragon fruit diseases in Thailand that are unsubstantiated and have not been correctly identified and scientifically confirmed. The lack of information on dragon fruit diseases in Thailand may contribute to a reduction in marketable yield. Knowledge of the precise diagnosis of disease and disease etiology is an essential step in appropriate disease management. During our investigation in 2020–2021, stem canker in red-fleshed dragon fruits was also observed in a cultivation farm in Phatthalung province, southern Thailand, where approximately 35% of plants were infected in the observed fields. There are some morphological variations and molecular properties indicating that this pathogen is a new species of *Neoscytalidium*. Therefore, this study aimed to isolate the pathogen causing stem canker in dragon fruit, to identify it based on morphological and molecular techniques and to assess the pathogenicity in *H. polyrhizus*.

## 2. Materials and Methods

### 2.1. Sample Collection and Fungal Isolation

Circular to irregularly shaped sunken orange cankers with a mass of pycnidia were observed in approximately 35% of dragon fruit plants cultivated in the observation area. Fruit and stem cankers were collected from a cultivation field in Phatthalung province, southern Thailand (7°45′24.2″ N 99°58′47.2″ E). A total of ten symptomatic samples were collected. The causal fungus was isolated using the tissue transplanting method [15,16]. The infected tissue samples were cut into small pieces (0.3 × 0.3 cm), and their surfaces were disinfected with 70% ethanol followed by 1% sodium hypochlorite (NaOCl). Excess NaOCl was removed using sterile distilled water (DW), which was repeated 3 times, and samples were dried on sterile Whatman© paper. The tissue samples were then placed on water agar (WA). The plates were then incubated at ambient temperature (28 ± 2 °C) for 3 days. Hyphal tips were cut and transferred to Petri dishes containing potato dextrose agar (PDA) to obtain the pure fungal strain. The fungal isolate was cultured on a PDA slant for further study.

### 2.2. Pathogenicity Test

To determine which fungi isolated from the infected samples caused stem canker in *H. polyrhizus*, the candidate fungal pathogens were cultured on PDA for 5 days, and the agar plug method was conducted [17,18,19]. The stems of *H. polyrhizus* were wounded using sterile fine needles, and agar plugs (0.5 cm) cut from 5-day-old PDA culture plates of each fungal isolate were placed directly onto the wounded stems. Agar plugs of PDA alone were used as the negative control. There were 3 replicates per treatment, and the inoculation was repeated three times. The inoculated plants were incubated in a moist box to maintain humidity at ambient temperature (28 ± 2 °C). The development of canker was observed for 7 days. 

### 2.3. Morphological Observation

The growth rate of fungal mycelia was measured on PDA. The macroscopic characteristics of pycnidia formation were observed [20] with a Leica S8AP0 stereomicroscope (Leica Microsystems, Wetzlar, Germany), whereas the microscopic characteristics of the conidia were examined with a Leica DM750 light microscope (Leica Microsystems, Wetzlar, Germany). The dimensions of the conidia were measured. The fungal cultures were then deposited in a culture collection, and the accession number was assigned.

### 2.4. Molecular Identification

The PCR amplification of the internal transcribed spacer (ITS), translation elongation factor 1-α (*tef1-α*) and *β*-tubulin (*tub*) gene regions was conducted using the primer pair ITS1/ITS4 [21] for ITS, EF1-728F [22]/EF2 [23] for the *tef1-α* gene and T1/Bt2b [24,25] for the *tub* gene. For DNA extraction, fungal isolates were cultured on PDA for 2 days and subjected to DNA extraction using the mini-preparation method according to the method of Saitoh et al. [26]. The PCR reaction mixtures contained a DNA template, 10 pmol of each primer, 2× Green PCR Master Mix (Thermo Scientific, Waltham, Massachusetts, United States) and distilled water (DW) in a 50 μL tube. PCR was performed with BIO-RAD T100^TM^ Thermal Cycler (Hercules, CA, USA), and the PCR conditions were 3 min at 95 °C, followed by 35 cycles of 95 °C for 30 s, 50 °C for 30 s, 72 °C for 1 min and the final extension at 72 °C for 10 min. The PCR products were visualized with agarose gel electrophoresis. 

The PCR products were then sequenced by the Ward Medic Ltd. sequencing service (Bangkok, Thailand) using the same primers that were used in the PCR reaction. The phylogenetic trees were constructed with related species acquired from GenBank (Table 1) with 1000 bootstrap replications. Sequences were aligned with Bioedit ver. 7.2 [27] using the ClustalW algorithm and ClustalX ver. 1.83 [28] and were manually adjusted as necessary. Phylogenetic tree estimation for each alignment was performed using maximum parsimony (MP), maximum likelihood (ML) and Bayesian inference (BI). MP trees were obtained using the heuristic search option with 1000 random additions of sequences and tree bisection and reconnection (TBR) as the branch-swapping algorithm of MEGA X [29]. ML trees were constructed using MEGA X based on the nearest neighbor interchange (NNI) as the heuristic method for tree inference, and 1000 bootstrap replicates were performed. Bayesian trees were generated using MrBayes ver. 3.2.7a [30]. Markov chain Monte Carlo (MCMC) runs were performed for 1,000,000 generations and sampled every 100th generation. The initial 25% of generations were discarded as burn-in, and the remaining trees were used to calculate the Bayesian inference posterior probability (BIPP) values. Phylogenetic trees were visualized using FigTree ver. 1.4.4.

## 3. Results

### 3.1. Disease Observation and Symptom Recognition

During sampling in 2020–2021, canker disease was detected in approximately 35% of *H. polyrhizus* in Phatthalung province. Visible external signs were small, circular, sunken, orange cankers on the stems and fruits of *H. polyrhizus* (Figure 1). The pycnidia formed on the canker surface contained black masses of pycnidial conidia (Figure 1C), and the cladodes subsequently rotted (Figure 1E,F). Three isolates of a *Scytalidium*-like fungus and five *Fusarium* isolates were consistently isolated from stem canker symptoms using the tissue transplanting method.

### 3.2. Pathogenicity Test

A total of eight isolates including three *Scytalidium*-like fungus and five *Fusarium* isolates were inoculated in cladodes of *H. polyrhizus*. The results showed that three isolates of *Scytalidium*-like fungus caused stem canker on *H. polyrhizus* cladodes similar to that observed in the field (Figure 2), whereas five isolates of *Fusarium* did not cause symptoms. The symptoms included orange flecks, the raising up of the infected site, and scab and pycnidia formation on the surface of dragon fruit cladodes (Figure 2). The fungi were re-isolated, and their morphology was matched with the original culture of the *Scytalidium*-like fungus.

### 3.3. Morphological Study

A total of three isolates of *Scytalidium*-like fungus that caused stem canker in *H. polyrhizus* were subjected to macroscopic and microscopic observations. From the macroscopic perspective, three isolates produced woolly colonies and olive green to grayish colonies on PDA (Figure 3). All three strains examined rapidly produced colonies, with an average growth rate of 3 cm/day, and they colonized the plate within 3 days. Mycelia were branched, septate, hyaline to brown and disarticulated into 0- to 2-septate arthroconidia singly or in arthric chains (Figure 3, Figure 4 and Figure 5). Arthroconidia varied in size in different isolates (*n* = 20) and were ellipsoid to ovoid, globose, oblong or cylindrical, singly or in arthric chains, 3.70–9.65 (6.24 ± 1.77) × 2.46–6.15 (3.73 ± 1.04) μm, with a length/width ratio of 1.67, as shown in Table 2. 

Pycnidia formed after incubation on PDA for 4 weeks, showing a black, irregularly shaped morphology, occurring singly or in aggregates (Figure 3). Pycnidial conidia were hyaline, ellipsoid, oblong or cylindrical and varied in size (*n* = 20) as 9.81–13.52 (11.48 ± 0.93) × 3.73–5.04 (4.27 ± 0.36) μm, with a length/width ratio of 2.68 (Figure 3 and Table 2). Based on the morphology observed in this study and the descriptions by Crous et al. [31], the fungus was identified as *Neoscytalidium* sp. The fungal isolates were then deposited in the culture collection of Pest Management Department, Faculty of Natural Resources, Prince of Songkla University, Thailand, with accession numbers PSU-HP01, TSU-HP01 and TSU-HP02 (Figure 3, Figure 4 and Figure 5).

**Table 2 jof-09-00197-t002:** Morphology comparison of *Neoscytalidium* spp. and related species.

Species	Colony	Arthroconidia	Pycnidia	Pycnidial Conidia	Reference
*Neoscytalidium dimidiatum*	White or hyaline colony, turned dark gray to black after 12 days on PDA	Cylindrical to round, hyaline to brown, singly or in arthric chains, 9.4 ± 1.2 × 5.1 ± 0.8 μm (length/width ratio = 1.85)	Produced after 3–4 weeks, black, irregularly shaped to ovoid, singly or in aggregates	Hyaline, ellipsoidal to ovoid, 11.1 ± 0.7 × 5.5 ± 0.5 μm (length/width ratio = 2.03)	[32]
*N. novaehollandiae*	–	Cylindrical, oblong to obtuse to doliiform, 0- to 1-septate, thick-walled, 5.2–11.2 × 2.9–4.9 μm, in arthric chains	Pycnidia (produced on pine needles on 1.5% water agar) semi-immersed or superficial with a mean diameter of 310 μm	Hyaline, ellipsoidal, becoming sepia, 0- to 1-septate, 3.1–4.1 × 12.2–13.8 μm	[33]
*N. hyalinum*	White with dense and hairy aerial mycelium and gradually turned gray to olive green	Truncated-cylindrical, oblong–obtuse to doliiform, hyaline to dark brown, 0- to 2-septate, in arthric chains, 4.57–9.85 × 3.125–27 μm	–	–	[34]
*N. orchidacearum*	Cream or white from above and reverse, with filamentous form or margin, flat	–	Stromatic, immersed, eventually erumpent, unilocular to multilocular (2~4-loculate), glabrous, brown to black, globose to subglobose, papillate, 200~500 μm diameter	Ellipsoidal to ovoid, hyaline, smooth, guttulate, aseptate becoming 2~3-septate, (10~) 12~13 (~15) × 3~5 (~6) μm (*n* = 50)	[31]
TSU-HP01	White, moderate growth of 3 cm/day, olive green to gray color and turned dark gray to black after 12 days	Ellipsoid to ovoid, globose, oblong–cylindrical, singly or in arthric chains, 0- to 2-septate, 3.70–9.65 (6.24 ± 1.77) × 2.46–6.15 (3.73 ± 1.04) μm, length/width ratio = 1.67	Rarely produced on PDA after 4 weeks of incubation, brown to dark brown, singly or in aggregates	Ellipsoid, oblong or cylindrical, 9.81–13.52 (11.48 ± 0.93) × 3.73–5.04 (4.27 ± 0.36) μm, length/width ratio = 2.68	This study

### 3.4. Phylogenetic Tree Analysis

The PCR products were approximately 510, 223 and 381 bp for the ITS, *tef1-α* and *tub* regions, respectively. A BLAST search in GenBank (The National Center of Biological Information; NCBI) showed that the sequences of the ITS, *tef1-α* and *tub* gene regions were more than 99% identical to *Neoscytalidium dimidiatum* for all isolates examined in this study. For the phylogenetic tree analysis, the dataset comprised 28 taxa, including *Guignardia bidwellii*, *G. citricarpa* and *G. philoprina* as the outgroup. The phylogenetic tree generated from the dataset consisted of 1114 characters, including the gap (ITS, 510; *tub*, 381; *tef1-α*, 223). The ML tree generated from the combined DNA sequences of ITS, *tub* and *tef1-α* of *Botryosphaeria*, *Cophinforma*, *Neofusicoccum* and *Neoscytalidium* is shown in Figure 6. Three isolates, PSU-HP01, TSU-HP01 and TSU-HP02, formed an independent clade with MP (77%) supports and were close to *N. dimidiatum*.

### 3.5. Taxonomy of a New Species

A total of three isolates representing most of the morphological variation presently recognized as *Neoscytalidium* [31] were subjected to DNA sequence analysis. Based on the morphological characteristics and DNA sequences of the ITS, *tef1-α* and *tub* regions, *Neoscytalidium* sp. isolates PSU-HP01, TSU-HP01 and TSU-HP02 were considered to be a new species (sp. nov.)

*Neoscytalidium hylocereum* S. Kheawleng, S. Intaraa-nun and M. Rodkaew, sp. nov., MycoBank 838004 (Figure 3).

Etymology: The name *hylocereum* refers to the host genus (*Hylocereus*).

Holotype: TSU-HP01.

Pathogen causing stem and fruit canker in *H. polyrhizus*. Sexual morph: Undetermined. Asexual morph: Hyphomycetous asexual morph not seen. Coelomycetous asexual morph: mycelium disarticulated into 0- to 2-septate, ellipsoid to ovoid, globose, oblong-cylindrical, singly or in arthric chains, 3.70–9.65 (6.24 ± 1.77) × 2.46–6.15 (3.73 ± 1.04) μm, length/width ratio = 1.67. Pycnidia rarely produced on PDA after 4 weeks of incubation, brown to dark brown, singly or in aggregates. Pycnidial conidia hyaline, ellipsoid, oblong or cylindrical and varied in size (*n* = 20) 9.81–13.52 (11.48 ± 0.93) × 3.73–5.04 (4.27 ± 0.36) μm, length/width ratio = 2.68.

Culture characteristics: Powdery white, fast-growing (3 cm/day), olive green to gray color and turned dark gray to black after 12 days.

Material examined: Thailand, Phatthalung province; on stem and fruit of dragon fruit (*H. polyrhizus*); August 3, 2020; S. Kheawleng, S. Intaraa-nun and M. Rodkaew (TSU-HP01 holotype). 

Notes: *Neoscytalidium hylocereum* PSU-HP01, TSU-HP01 and TSU-HP02 with the same DNA sequences were related to *N. dimidiatum* with 77% MP support, although without enough ML and PP support. Within the comparison of the tree regions of DNA sequences, there were two character differences in the ITS and *tef1-α* regions and four in the *tub* region. Therefore, *Neoscytalidium hylocereum* (TSU-HP01) is introduced as a new species.

## 4. Discussion

In this study, we identified the cause of canker in *H. polyrhizus* based on symptom recognition (Figure 1), pathogenicity testing (Figure 2), morphology (Figure 3, Figure 4 and Figure 5) and molecular analysis (Figure 6). For the pathogenicity test, the wounding method was successful in inoculating the fungal pathogen into the stem of *H. polyrhizus* (Figure 2). This result is in agreement with previous studies [35,36], where infection by a fungal pathogen was facilitated by wounding. The stem canker from the pathogenicity test appeared similar to that observed in the natural infection. However, the symptom found in the natural infection seemed more severe than that in the laboratory test, and this may have been due to the environmental conditions in the field favoring fungal germination and host infection. 

Stem canker in dragon fruit has been recognized to be caused by the fungus *N. dimidiatum* throughout Asian countries [10,14]; however, very few reports exist regarding the life cycle in dragon fruit. Infected dragon fruit shows three stages of symptoms, including sunken yellow to orange flecks, the raising up of the infected site due to tissue proliferation beneath the infected area and pycnidia forming on the surface of dragon fruit cladodes, which was observed in a new species, *N. hylocereum*, in this study. The fungus produced two types of conidia: arthroconidia, which were formed by the breaking up of individual cells and clusters of cells of mature hyphae, and pycnidial conidia, which formed in pycnidia embedded in the surface of the host [32]. Our result is in agreement with a previous report indicating that *N. hylocereum* produced two types of conidia on host surfaces.

The identification of the Botryosphaeriaceae species is commonly based on the morphology of the anamorph [37]. The most abundant characteristic of fungi in the genus *Neoscytalidium* is that the hyphae are branched, septate, brown and disarticulated into 0- to 1-septate arthrospores [31]. In this study, three isolates of *Neoscytalidium* spp. displayed these morphological characteristics (Figure 3, Figure 4 and Figure 5). The morphology of three isolates (PSU-HP01, TSU-HP01 and TSU-HP02) was closely related to *N. dimidiatum* (Table 2), but there were some distinct morphological characteristics, and DNA sequence analysis supported the classification of the three isolates found in this study as a new species. For instance, the average size of *N. dimidiatum* arthroconidia was 9.4 × 5.1 μm with a length/width ratio of 1.85 [32]; however, the arthroconidia of TSU-HP01 were smaller than those of *N. dimidiatum*, at 4.82 × 2.74 μm (ratio = 1.75) and 6.24 × 3.73 μm (length/width ratio = 1.67), respectively, as described in Table 2. The pycnidia of *N. dimidiatum* were black, irregularly shaped to ovoid, singly or in aggregates and were produced after 3–4 weeks [32], whereas the pycnidia of TSU-HP01 were rarely produced on PDA (Table 2). Furthermore, the pycnidial conidia of *N. dimidiatum* were hyaline, ellipsoidal to ovoid, with an average size of 11.1 × 5.5 μm, and a length/width ratio of 2.3 [32], whereas the pycnidial conidia of TSU-HP01 were ellipsoid, oblong or cylindrical, with an average size of 11.48 × 4.27 μm and a length/width ratio of 2.68 (Table 2). 

Based on our results, ML trees indicated that the isolates of *Neoscytalidium* in this study were morphologically and genetically distinct from other isolates (Figure 6 and Table 2). Although the DNA sequences of *Neoscytalidium* sp. isolates PSU-HP01, TSU-HP01 and TSU-HP02 showed sequence similarity to *N. dimidiatum* of more than 99% according to BLAST search analysis (NCBI), the phylogenetic tree of combined DNA sequences revealed a new clade of *Neoscytalidium* spp. found in this study (Figure 6). The combined DNA sequences of the ITS, *tef1-α* and *tub* gene regions placed these three isolates in a different clade with respect to *N. dimidiatum* (Figure 6), with 77% MP support. Based on the morphological characteristics and DNA sequences, the three isolates of *Neoscytalidium* spp. found in this study were considered a new species (sp. nov.). The biota of the new species was deposited in Mycobank with accession number 838004 for *N*. *hylocereum*. 

Fungi of the genus *Neoscytalidium* have been found to cause disease in several plant species, with worldwide distribution. For instance, *N. dimidiatum* has been documented to cause wood canker in grapevines in California, USA [38]; dieback in *Ficus benjamina* L. in Mexico [39]; canker, shoot blight and fruit rot in almond trees in California, USA [32]; dieback, shoot blight and branch canker in willow trees in Turkey [40]; and shoot blight, dieback and canker in apricot trees in Turkey [41]. This fungal species has been reported to cause stem canker in pitaya (*H. undatus* and *H. polyrhizus*) in Taiwan [42], stem and fruit canker in *H. undatus* in Florida [43] as well as canker disease in pitaya in Hainan, China [44]. 

Currently, only four species of the genus *Neoscytalidium* have been confirmed to cause disease in several plants worldwide. In southeastern Malaysia, stem canker in red-fleshed dragon fruit (*H. polyrhizus*) was reported to be caused by *N. dimidiatum* [10]. However, the results of the present study reveal that the causal agent of canker in red-fleshed dragon fruit (*H. polyrhizus*) in Thailand seems to be a new species, namely, *N. hylocereum*. Therefore, to our knowledge, this is the first report of the new species, *N. hylocereum*, causing canker in *H. polyrhizus* in Thailand and elsewhere.

## 5. Conclusions

The majority of the species of *Neoscytalidium* have been reported worldwide, and only four species have been found to cause disease in several plant species. In this study, we introduced a new species derived from *H. polyrhizus* in southern Thailand, by means of geographical records, pathogenicity testing, and morphological and phylogenetic evidence. Until now, only two species of *Neoscytalidium* (*N. dimidiatum* and *N. orchidacearum*) have been reported in Thailand [13,31]. Over the years, there has been a lack of research on *Neoscytalidium* fungi in Thailand. Therefore, the introduction of *N. hylocereum* sp. nov. enriches the literature regarding the diversity study of *Neoscytalidium* fungi in Thailand. Knowledge of the precise diagnosis of plant diseases is an essential step in disease management, and further study and verification regarding disease control is required in the near future.

## Figures and Tables

**Figure 1 jof-09-00197-f001:**
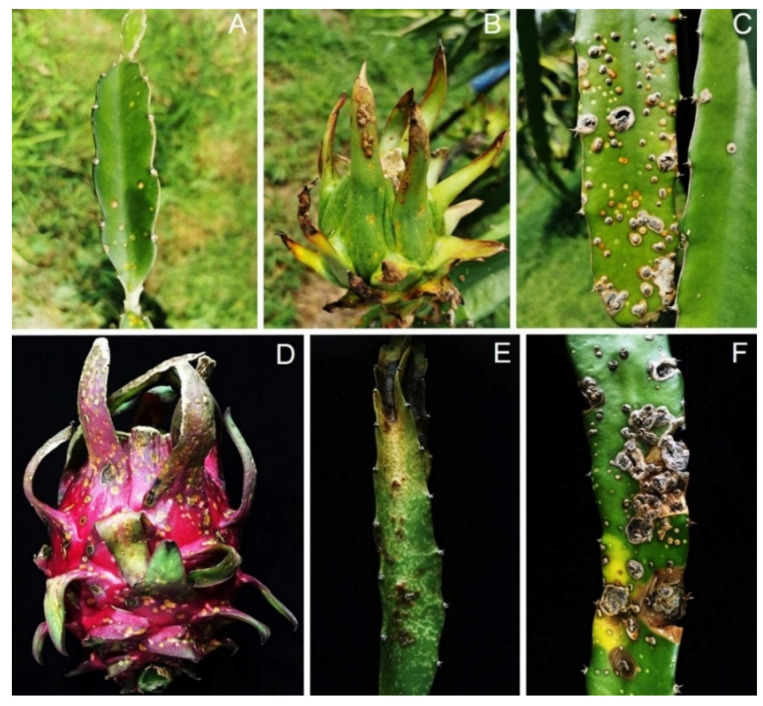
Canker of *Hylocereus polyrhizus* observed in the field: circular sunken orange cankers on cladode (**A**) and fruit (**B**); black pycnidia formed on the surface of cankers on cladodes (**C**) and fruit (**D**); and infected tissues that became rotted (**E**,**F**).

**Figure 2 jof-09-00197-f002:**
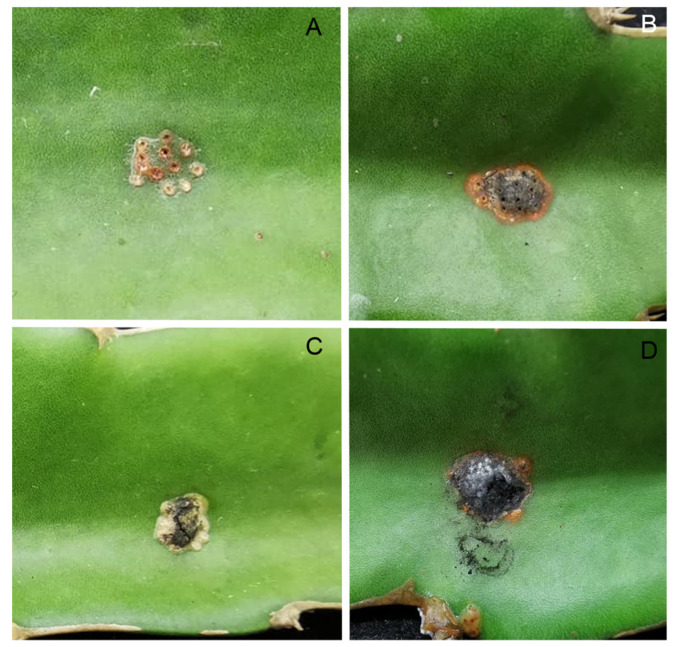
Pathogenicity test of three *Neoscytalidium* sp. isolates by means of agar plug method on wounded *H*. *polyrhizus* cladodes: control (**A**), PSU-HP01 (**B**), TSU-HP01 (**C**) and TSU-HP02 (**D**).

**Figure 3 jof-09-00197-f003:**
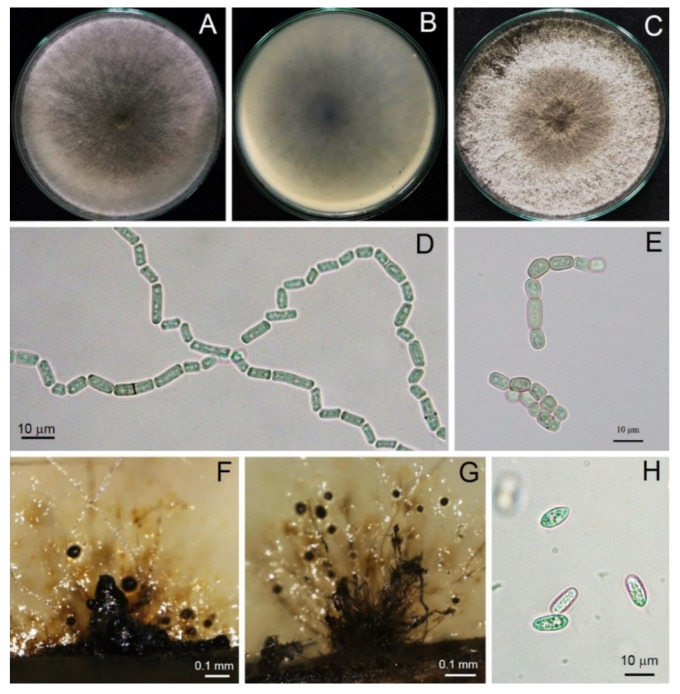
Morphological characteristics of *Neoscytalidium* sp. (TSU-HP01): three-day-old colony on PDA—top (**A**) and bottom views (**B**); one-week-old colony on PDA (**C**); arthroconidia ((**D**,**E**); ×400); black pycnidia ((**F**); ×100); pycnidia and conidiogenous cells ((**G**); ×100); pycnidial conidia ((**H**); ×400).

**Figure 4 jof-09-00197-f004:**
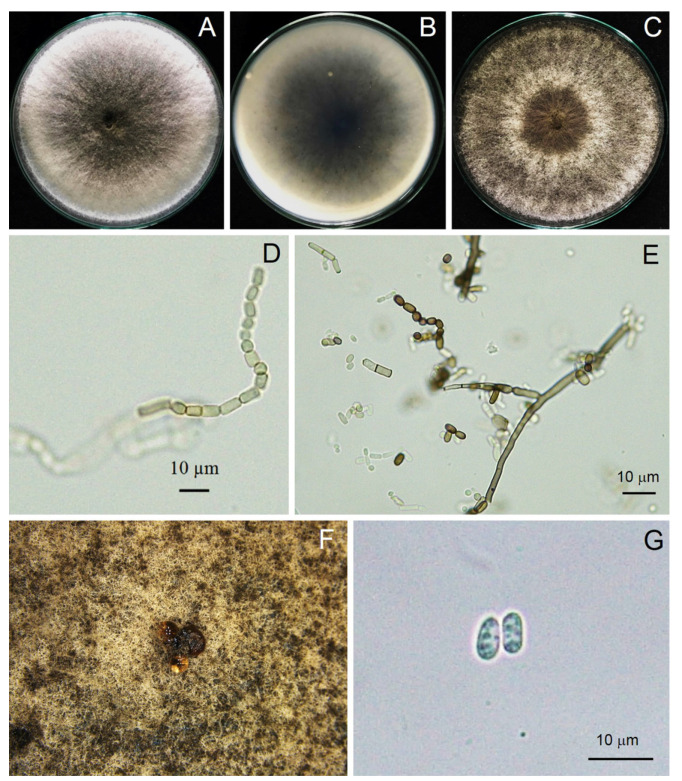
Morphological characteristics of *Neoscytalidium* sp. (PSU-HP01): three-day-old colony on PDA—top (**A**) and bottom views (**B**); one-week-old colony on PDA (**C**); arthroconidia ((**D**,**E**); ×400); black pycnidia (**F**); pycnidial conidia ((**G**); ×400).

**Figure 5 jof-09-00197-f005:**
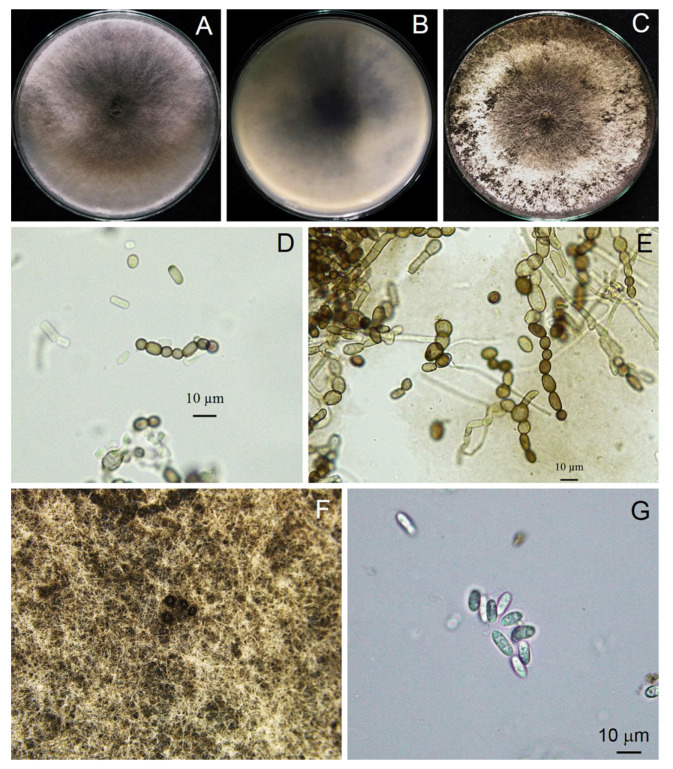
Morphological characteristics of *Neoscytalidium* sp. (TSU-HP02): three-day-old colony on PDA—top (**A**) and bottom views (**B**); one-week-old colony on PDA (**C**); arthroconidia ((**D**,**E**); ×400); black pycnidia (**F**); and pycnidial conidia ((**G**); ×400).

**Figure 6 jof-09-00197-f006:**
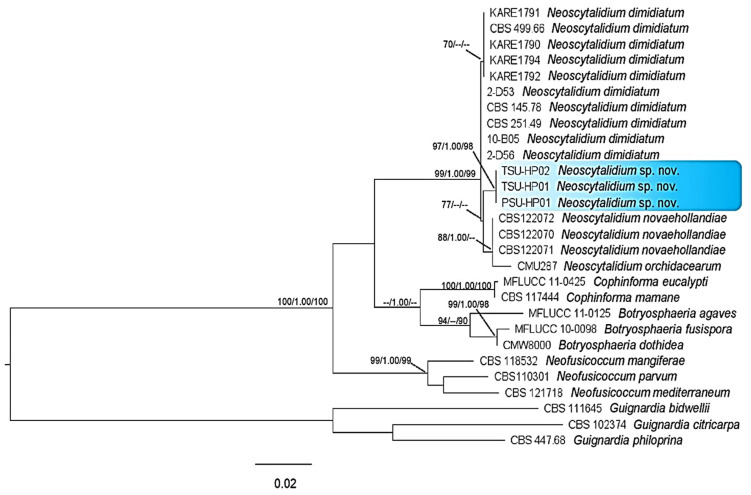
Phylogenetic tree of combined DNA sequences of ITS, *tef1-α* and *tub* of *Neoscytalidium* sp. and related species; PSU-HP01, TSU-HP01 and TSU-HP02 were samples from this study. Bootstrap support values for MP and ML equal to or greater than 70% and Bayesian inference (BI) equal to or greater than 0.90 were defined as MP/BI/ML above or below the nodes. *Guignardia bidwellii*, *G. citricarpa* and *G. philoprina* were used as the outgroup.

**Table 1 jof-09-00197-t001:** Collection details and GenBank accession numbers of isolates included in this study.

Species	Voucher/Culture	Location	Host	GenBank Accession Number		
				ITS	*tef1-α*	*tub*
*Botryosphaeria agaves*	MFLUCC 11-0125 ^T^	Thailand	*Agaves* sp.	JX646791	JX646856	JX646841
*B. fusispora*	MFLUCC 10-0098 ^T^	Thailand	*Caryota* sp.	JX646789	JX646854	JX646839
*B. dothidea*	CMW8000	Switzerland	*Prunus* sp.	AY236949	AY236898	AY236927
*Cophinforma eucalypti*	MFLUCC 11-0425	Thailand	*Eucalyptus* sp.	JX646800	JX646865	JX646848
*C. mamane*	CBS 117444	Venezuela	*Eucalyptus* sp.	KF531822	KF531801	KF531802
*Guignardia bidwellii*	CBS 111645	USA: Missouri	*Parthenocissus quinquefolia*	FJ824766	FJ824772	FJ824777
*G. citricarpa*	CBS 102374 ^T^	Brazil	*Citrus aurantium*	FJ824767	FJ538371	FJ824778
*G. philoprina*	CBS 447.68	USA	*Taxus baccata*	FJ824768	FJ824773	FJ824779
*Neofusicoccum mangiferae*	CBS 118532	Australia	*M. indica*	AY615186	DQ093220	AY615173
*N. mediterraneum*	CBS 121718 ^T^	Greece	*Eucalyptus* sp.	GU251176	GU251308	GU251836
*N. parvum*	CBS 110301	Portugal	*Vitis vinifera*	AY259098	AY573221	EU673095
*Neoscytalidium dimidiatum*	10-B05	USA: Kern	*P. dulcis*	MG021589	MG021541	MG021486
	KARE1790	USA: Madera	*P. dulcis*	MG021578	MG021567	MF991145
	KARE1791	USA: Madera	*P. dulcis*	MG021579	MG021531	MG021476
	KARE1792	USA: Madera	*P. dulcis*	MG021580	MG021532	MG021477
	KARE1794	USA: Madera	*P. dulcis*	MG021582	MG021534	MG021479
	2-D53	USA: Madera	*Ficus carica*	MG021568	MG021521	MG021511
	2-D56	USA: Fresno	*F. carica*	MG021570	MG021523	MG021513
	CBS 145.78 ^I,^*	United Kingdom	*Homo sapiens*	KF531816	KF531795	KF531796
	CBS 499.66 *	Mali	*Mangifera indica*	KF531820	KF531798	KF531800
	CBS 251.49 *	USA	*Juglans regia*	KF531819	KF531797	KF531799
*N. novaehollandiae*	CBS122071 ^T^	Australia	*Crotalaria medcaginea*	EF585540	EF585580	MT592760
	CBS122070	Australia	*Grevillea agrifolia*	EF585539	EF585579	MT592759
	CBS122072	Australia	*Adansonia gibbosa*	EF585535	EF585581	MT592761
*N. orchidacearum*	CMU287 ^T^	Thailand	*Cattleya* sp.	KY933091	N/A	N/A
***Neoscytalidium* sp.**	**PSU-HP01**	**Thailand**	** *Hylocereus polyrhizus* **	**LC590859**	**LC590862**	**LC647832**
***Neoscytalidium* sp.**	**TSU-HP01**	**Thailand**	** *H. polyrhizus* **	**LC590860**	**LC590863**	**LC647833**
***Neoscytalidium* sp.**	**TSU-HP02**	**Thailand**	** *H. polyrhizus* **	**LC590861**	**LC590864**	**LC647834**

^T^ = ex-type strain; ^I^ = ex-isotype strain; N/A = not available. * Name changed from *Neoscytalidium hyalinum*; isolates from this study are indicated in bold.

## Data Availability

Not applicable.

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
