# Peer review of "A New Species of Neoscytalidium hylocereum sp. nov. Causing Canker on Red-Fleshed Dragon Fruit (Hylocereus polyrhizus) in Southern Thailand"

_jof, 2023, doi:10.3390/jof9020197_

Round 1

Reviewer 1 Report

This article presented A New Species of Neoscytalidium hylocereum sp. nov. Causing Canker on Red-Fleshed Dragon Fruit (Hylocereus polyrhizus) in Southern Thailand. The study is well organized and data is well arranged. The findings would be helpful for future studies. Before recommending this article for publication, there are some shortcomings for that should be resolve.

Overall, the study is well designed and presented in a good way, but mostly the literature is not cited. Grammatical and typos must be revised.

In abstract the authors should add brief details of morphological characters which were study.

All methods must be mentioned in the abstract which used to identify the new species.

Also specific results of pathogenicity test must be mention.

In introduction first paragraph economic and medicinal importance of the Dragon fruit in detail must be mention.

Also discuss the details of canker and the mechanism of the fungus studied in this article.

How much devastation caused by N. hylocereum.

Section 2.2 and section 2.3 must be cited with relevant studies.

https://doi.org/10.1016/j.bcab.2020.101729, https://doi.org/10.1002/aoc.5190,

The authors should mention at which resolution the fungus mycelia was observed.

Figures of pathogenicity test must be more clarify.

Add future perspective of the study in conclusion.

Author Response

Reviewer 1

This article presented A New Species of Neoscytalidium hylocereum sp. nov. Causing Canker on Red-Fleshed Dragon Fruit (Hylocereus polyrhizus) in Southern Thailand. The study is well organized and data is well arranged. The findings would be helpful for future studies. Before recommending this article for publication, there are some shortcomings for that should be resolve.

Overall, the study is well designed and presented in a good way, but mostly the literature is not cited. Grammatical and typos must be revised.

In abstract the authors should add brief details of morphological characters which were study.

Answer: We have added in abstract.

All methods must be mentioned in the abstract which used to identify the new species.

Answer: We have added in abstract.

Also specific results of pathogenicity test must be mention.

Answer: We have added in abstract.

In introduction first paragraph economic and medicinal importance of the Dragon fruit in detail must be mention.

Answer: We have mentioned in the first paragraph of introduction.

Also discuss the details of canker and the mechanism of the fungus studied in this article.

Answer: We have added in discussion part.

How much devastation caused by N. hylocereum.

Answer: We have added in last paragraphed of introduction.

Section 2.2 and section 2.3 must be cited with relevant studies.

https://doi.org/10.1016/j.bcab.2020.101729, https://doi.org/10.1002/aoc.5190,

Answer: We have added in cited and references.

The authors should mention at which resolution the fungus mycelia was observed.

Answer: We have added in the figures.

Figures of pathogenicity test must be more clarify.

Answer: We have removed and placed the new one.

Add future perspective of the study in conclusion.

Answer: We have added perspective in conclusion.

Reviewer 2 Report

The manuscript (A New Species of Neoscytalidium hylocereum sp. nov. Causing Canker on Red-Fleshed Dragon Fruit (Hylocereus polyrhizus) in Southern Thailand) explores a new fungal disease affecting the production of dragon fruits.

I think the study is more suitable to be a short communication.

This is not the first report of the pathogen Neoscytalidium sp. on dragon fruit please check the following paper (Identification and Molecular Characterizations of Neoscytalidium dimidiatum Causing Stem Canker of Red-fleshed Dragon Fruit (Hylocereus polyrhizus) in Malaysia). Hence, I suggest to compare between Neoscytalidium hylocereum and Neoscytalidium dimidiatum in different aspects.

Lines 38-40 I wonder why the authors did not refer to the pathogen Neoscytalidium dimidiatum?

Line 60 Please explain the symptoms in the field?

Did you evaluate disease severity or incidence of the symptoms in the field?

Line 63 The samples should be washed for at least three times with sterile distilled water after sodium hypochloride treatment.

Line 66 The incubation temperature is a little high (28 ± 2°C).

Line 76 Please add humidity percentage.

Line 84 Please add to the materials and methods how you investigate macroscopic and microscopic characteristics.

Line 96 Please provide an image for PCR products.

I think you should evaluate the disease in the pathogenicity test, you can use the disease rating scale to compare the severity of the isolates.

Most of the results of the pathogenicity test were not provided (data not shown). I ask the authors to provide the figures and tables as supplementary files.

Figure 3 explains only the isolate TSU-HP01, I suggest you add more two figures for the other two strains of the pathogen.

The conclusion is weak, please rewrite the conclusion.

The English of the manuscript must be improved. Some words are not suitable such as faced which should be replaced with infected. I recommend that the authors should ask English editing service.

Author Response

Reviewer 2

The manuscript (A New Species of Neoscytalidium hylocereum sp. nov. Causing Canker on Red-Fleshed Dragon Fruit (Hylocereus polyrhizus) in Southern Thailand) explores a new fungal disease affecting the production of dragon fruits.

I think the study is more suitable to be a short communication.

Answer: According to a novel finding of new species we prefer this manuscript as a research article.

This is not the first report of the pathogen Neoscytalidium sp. on dragon fruit please check the following paper (Identification and Molecular Characterizations of Neoscytalidium dimidiatum Causing Stem Canker of Red-fleshed Dragon Fruit (Hylocereus polyrhizus) in Malaysia).

Answer: Yes, we have mentioned about N. dimidiatum in introduction and in discussion parts.

Hence, I suggest to compare between Neoscytalidium hylocereum and Neoscytalidium dimidiatum in different aspects.

Answer: As showed in table 2, we compared morphology (colony, dimension of conidia, pycnidial conidia) of a new species with related species including N. dimidiatum.

Lines 38-40 I wonder why the authors did not refer to the pathogen Neoscytalidium dimidiatum?

Answer: We have added Neoscytalidium dimidiatum into this paragraph.

Line 60 Please explain the symptoms in the field?

Answer: We have added symptom in the field.

Did you evaluate disease severity or incidence of the symptoms in the field?

Answer: We have evaluated infected plants, approximately 35% of dragon fruits showed stem canker.

Line 63 The samples should be washed for at least three times with sterile distilled water after sodium hypochloride treatment.

Answer: Yes, after disinfected with NaOCl, samples were washed with sterile distilled water for 3 times.

Line 66 The incubation temperature is a little high (28 ± 2°C).

Answer: According to temperature in southern part of Thailand, so the incubation temperature is a little high.

Line 76 Please add humidity percentage.

Answer: We have added humidity percentage.

Line 84 Please add to the materials and methods how you investigate macroscopic and microscopic characteristics.

Answer: We have added ‘Dimensions of fungal conidia were measured”.

Line 96 Please provide an image for PCR products.

Answer: We did not take the picture of PCR products.

I think you should evaluate the disease in the pathogenicity test, you can use the disease rating scale to compare the severity of the isolates.

Answer: Because this study we used point inoculate with and without pathogen and disease progress was qualify observed from inoculated point, therefore we did not rate the scale.

Most of the results of the pathogenicity test were not provided (data not shown). I ask the authors to provide the figures and tables as supplementary files.

Answer: Pathogenicity test result was provided in Fig. 2. For fungi that recovered from inoculated dragon fruit were not showed in this study because the result showed similar result as observed in Fig 3–5.

Figure 3 explains only the isolate TSU-HP01, I suggest you add more two figures for the other two strains of the pathogen.

Answer: Actually, TSU-HP01 is used as sample for explanation in this study due to all 3 isolates showed the same morphology. However, we have already added 3 isolates in this manuscript.

The conclusion is weak, please rewrite the conclusion.

Answer: We have re-written the conclusion.

The English of the manuscript must be improved. Some words are not suitable such as faced which should be replaced with infected. I recommend that the authors should ask English editing service.

Answer: We have used “infected” instead of “faced”. This manuscript has been edited by MDPI English editing service with certificate.

Round 2

Reviewer 2 Report

I do think the disease severity and incidence are important and missing in this study because the data in the pathogenicity test is not enough. Additionally, I do think that the authors should provide an image of PCR products.

Author Response

Reviewer 2

I do think the disease severity and incidence are important and missing in this study because the data in the pathogenicity test is not enough. Additionally, I do think that the authors should provide an image of PCR products.

Answer: For pathogenicity test, we have tested ability of fungal isolates to cause disease on dragon fruit cladodes, which resulted as 1) infected with symptom and 2) not infect without symptom, which considered as quality study. Therefore, we did not score for severity and incidence in this study.

We have checked PCR products after PCR amplification by gel electrophoresis and sent to DNA sequencing. Therefore, we don’t have image of PCR products but we have added the product size of PCR of each DNA region into text.  
